# Mining the neuroimaging literature

**Jérome Dockès[1][†], Kendra M Oudyk[2]\*[†], Mohammad Torabi[2], Alejandro I de la Vega[3], Jean-Baptiste Poline[2]\***

[1]National Institute for Research in Digital Science and Technology (INRIA), Paris, France; [2]Montreal Neurological Institute, McGill University, Montreal, Canada; [3]The University of Texas at Austin, Austin, United States

## eLife assessment

The study presents an **important** ecosystem designed to support literature mining in biomedical research, showcasing a methodological framework that includes tools like Pubget for article collection and labelbuddy for text annotation. The **solid** evidence presented for these tools suggests they could streamline the analysis and annotation of scientific literature, potentially benefiting research across a range of biomedical disciplines. While the primary focus is on neuroimaging literature, the applicability of these methods and tools might extend further, offering an advance in the practices of meta-research and literature mining.

**\*For correspondence:**
kendra.oudyk@gmail.com (KMO);
jean-baptiste.poline@mcgill.ca (J-BP)

[†]These authors contributed equally to this work

**Competing interest:** The authors declare that no competing interests exist.

**Abstract** Automated analysis of the biomedical literature (*literature mining*) offers a rich source of insights. However, such analysis requires collecting a large number of articles and extracting and processing their content. This task is often prohibitively difficult and time-consuming. Here, we provide tools to easily collect, process, and annotate the biomedical literature. In particular, https://neuroquery.github.io/pubget/pubget.html is an efficient and reliable command-line tool for downloading articles in bulk from PubMed Central, extracting their contents and metadata into convenient formats, and extracting and analyzing information such as stereotactic brain coordinates. https://jeromedockes.github.io/labelbuddy/labelbuddy/current/ is a lightweight local application for annotating text, which facilitates the extraction of complex information or the creation of ground-truth labels to validate automated information extraction methods. Further, we describe repositories where researchers can share their analysis code and their manual annotations in a format that facilitates reuse. These resources can help streamline text mining and meta-science projects and make text mining of the biomedical literature more accessible, effective, and reproducible. We describe a typical workflow based on these tools and illustrate it with several example projects.

## Introduction
### The need for literature mining is growing fast

More than 1 million papers are indexed by PubMed each year, about two papers per minute (*Landhuis, 2016*). This vast and fast-growing collection of knowledge has the potential to accelerate scientific insight and discovery. However, this information is represented as unstructured natural language texts across heterogeneous sources, making it difficult for researchers to access and analyze. As the number of publications continues to grow, there is an increasing need for tools and frameworks to systematically extract information from the scientific literature.

To understand the state of science, researchers and practitioners rely on syntheses in the form of reviews, meta-analyses, and other types of meta-research (i.e., research on research). These can be thought of as forms of 'literature mining', which is a special form of text-mining where the goal is to extract information from the scientific literature. This type of research requires finding and collecting

articles, systematically annotating information, and analyzing the resulting data. However, this painstaking work is traditionally performed manually, making it difficult to scale to the fast-growing literature. This process can be facilitated with user-friendly tools, but manual work is not easily reproduced. This is important because there is growing interest in continuously updating summaries of the literature (*Simmonds, 2022*), as well as growing concern over reproducibility of science in general (*Baker, 2016*).

Advances in natural language processing promise to automate many aspects of literature mining, potentially increasing scalability and reproducibility of meta-research. For example, specialized biomedical text-mining models are able to automatically recognize key entities in natural language, such as proteins, genes, and drugs (for a review, see *Huang et al., 2020*). More recently, there has been a growing interest in domain general large language models, which promise to perform well on a variety of language tasks, including scientific literature mining (e.g., *Tinn et al., 2023*).

However, even as we move toward more highly automated literature mining, there is still a need for manual work: we need ground-truth labels for training and evaluating these automated methods. Further, these projects require large datasets, which are difficult to collect and curate. This is more of a software engineering task than a research task, for which researchers are generally not well-prepared. Thus, right now we need tools to automate as much of the literature mining process as we can, while making the remaining manual steps as easy and efficient as possible.

Here, we present *litmining*, an ecosystem to optimize the scalability, reproducibility, and accessibility of literature mining tasks, from collecting documents, to transforming them into usable formats, to manually and automatically extracting information. We offer a flexible environment that can be used for projects that have various needs. To illustrate this, we describe a variety of projects that use our ecosystem, from meta-analysis, to model evaluation, to automated and manual meta-research projects. In what follows, we will further expand on current approaches to literature mining, outline problems with those approaches, and describe our proposed workflow and set of tools. While this article focuses on the neuroimaging field, we demonstrate that our ecosystem may be used in a variety of contexts.

## Limitations of current approaches to literature data collection

We can divide approaches to literature data collection into three broad categories, each with its own disadvantages in terms accessibility, reproducibility, and scalability.

The first, simplest approach is to rely on an existing and open corpus of published articles and extracted information. In neuroimaging, there are two such corpora: NeuroSynth (*Yarkoni et al., 2011*) and NeuroQuery (*Dockès et al., 2020*). They have been used extensively (especially NeuroSynth), reflecting the interest in using this type of data. However, these static article collections are limited in

| | Existing workflows for collecting literature-mining data | | | Using our *litmining* ecosystem |
|---|---|---|---|---|
| | **1) Re-using** an existing corpus | **2) Manually** collecting papers | **3) Automatically** collecting papers | |
| **Accessible** (low technical expertise needed) | medium | high | low | medium |
| **Scalable** (not time consuming) | low | low | high | high |
| **Reproducible** | low (dataset) – high (analysis) | low | medium | high |

**Figure 1.** Advantages and disadvantages of different approaches to data collection for literature mining. The first two approaches are not scalable, while the third is not accessible to researchers with lower technical expertise. We have aimed to make our approach as scalable, reproducible, and accessible as possible.

their coverage of the literature over time and topics, and they do not provide the full texts of papers, hence not suitable for many types of meta-research projects.

A second approach is to manually collect articles and extract information. In this case, a researcher typically performs a search on PubMed, downloads PDF files or reads the articles online, and then enters the relevant information into a spreadsheet. This is the most accessible approach, but it is a massive undertaking when many articles are involved; it is not scalable. Moreover, it is challenging to report this manual process completely and transparently, presenting a barrier to reproducibility.

The third and most scalable approach is to automate the data collection and curation. This involves writing and validating software to query a web repository's Application Programmer Interface (API), download articles, parse the output, extract the metadata and content, and prepare the text for annotation and analysis. This often represents a prohibitive amount of work, especially if this step is to be reproducible. Redoing this work for each new project is an extremely inefficient use of time and energy for the research community in addition to being error-prone and variably reproducible.

*Figure 1* summarizes the advantages and disadvantages of these three approaches, as well as our approach.

## Our proposed workflow and ecosystem

In contrast to these workflows, we present both an ecosystem of well-tested tools and repositories, as well as a set of recommendations for easier, more scalable, and more reproducible literature-mining and meta-research projects. *Figure 2* shows our proposed open ecosystem and workflow for literature mining projects.

For the first step of collecting documents and extracting content, we introduce https://neuro-query.github.io/pubget/pubget.html. This tool downloads open-access articles from PubMed Central (*National Library of Medicine, 2003*) and extracts their metadata and text. Further, for neuroimaging articles, it extracts the stereotactic coordinates of results in the three-dimensional space of the brain. The extracted information is stored in a way that facilitates its subsequent use and increases interoperability with other tools, such as the NiMARE library for neuroimaging meta-analysis (*Salo and Bottenhorn, 2018*).

To extract information from papers, we introduce a text-annotation software, https://jeromed-ockes.github.io/labelbuddy/labelbuddy/current/. This tool is critical as many meta-research projects require some manual annotation, for example, to provide validation data for an automatic extraction algorithm, or when automation is not feasible because the corpus is too small. While there already are existing tools for annotation, most are meant for a large team of annotators and generally not adapted to the annotation of research articles. labelbuddy is a lightweight solution and is easy to integrate in any workflow, making manual annotations as reproducible and scalable as possible, as well as interoperable with other tools in the *litmining* ecosystem (such as pubget, see below).

When information may be extracted automatically, without the need for manual annotation, pubget is easily extensible through plugins, allowing extension with additional automated processing steps. To illustrate this, we also provide https://github.com/litmining/pubextract, a collection of pubget plugins for extracting additional information from downloaded articles. Currently, pubextract offers functions for automatically extracting sample sizes, participant demographics, author locations, and author genders.

We further provide labelbuddy-annotations, an open repository of annotations (created with label-buddy on documents collected with pubget) that can be verified and reused across projects and updated over time.

We have created a landing page for all these tools and repositories at https://litmining.github.io/, which explains the basic workflow in text and instructional videos.

## Goals and scope

While researchers have a clear idea of the analyses they want to perform, the vast majority of time and energy is spent on the data collection and curation steps. We hope that the tools and recommendations we present here will make this process easier, more scalable, and more reproducible, and will therefore facilitate meta-research and literature mining.

In what follows, we illustrate how our flexible set of tools can be used to perform a variety of meta-research tasks. We highlight projects that demonstrate the use of the ecosystem for reproducing or

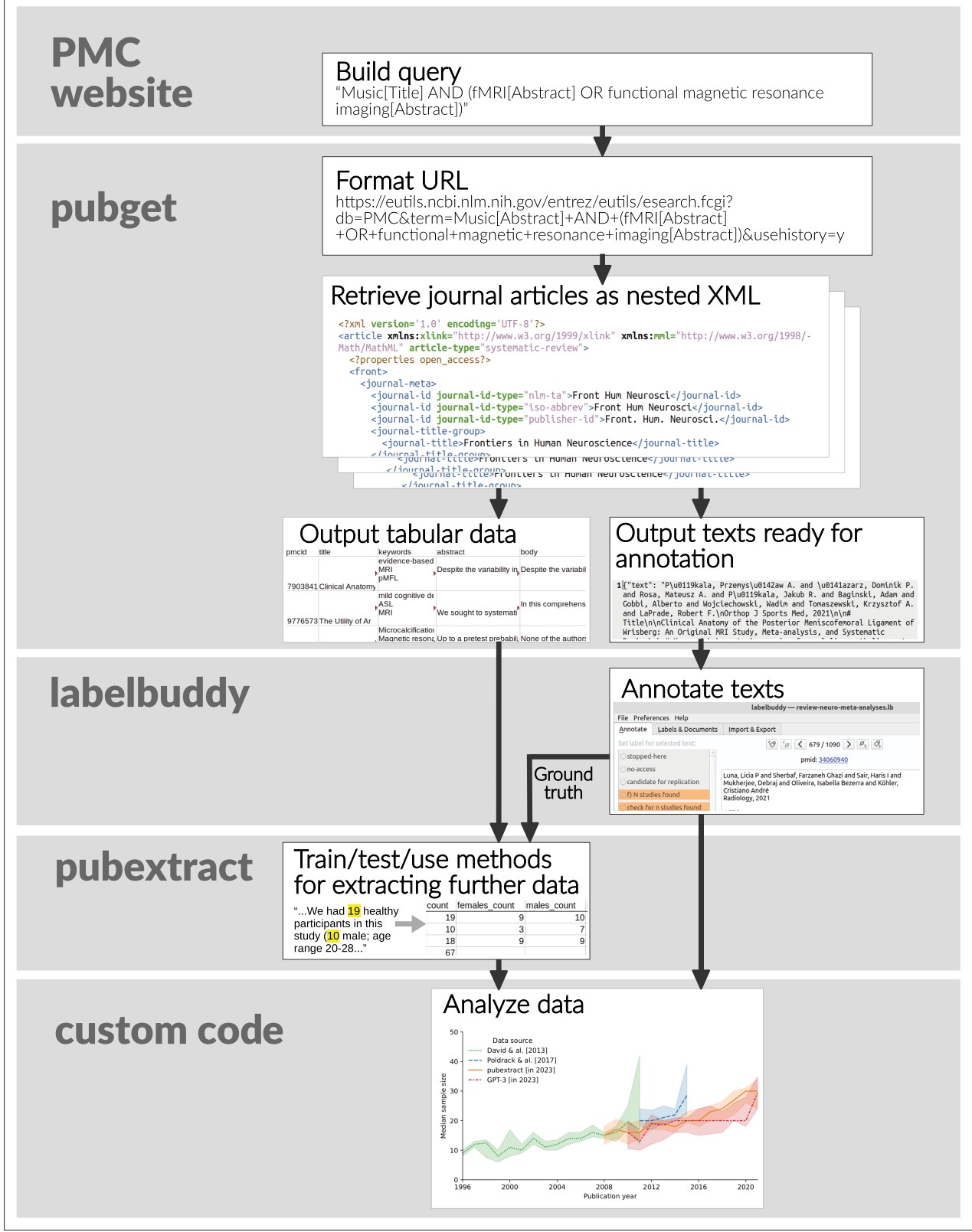

**Figure 2.** Our suggested workflow and litmining ecosystem of tools for efficient, reproducible meta-research. Our tool pubget performs the tasks of collecting documents and extracting content. Our tools labelbuddy, pubget, and pubextract can be used to manually and automatically extract information. We have an open repository of labelbuddy annotations, where researchers can re-use, update, and add new annotations. For the step of analyzing the data, each project would have its own code, which we hope would be tracked and shared in its own repository on GitHub or elsewhere.

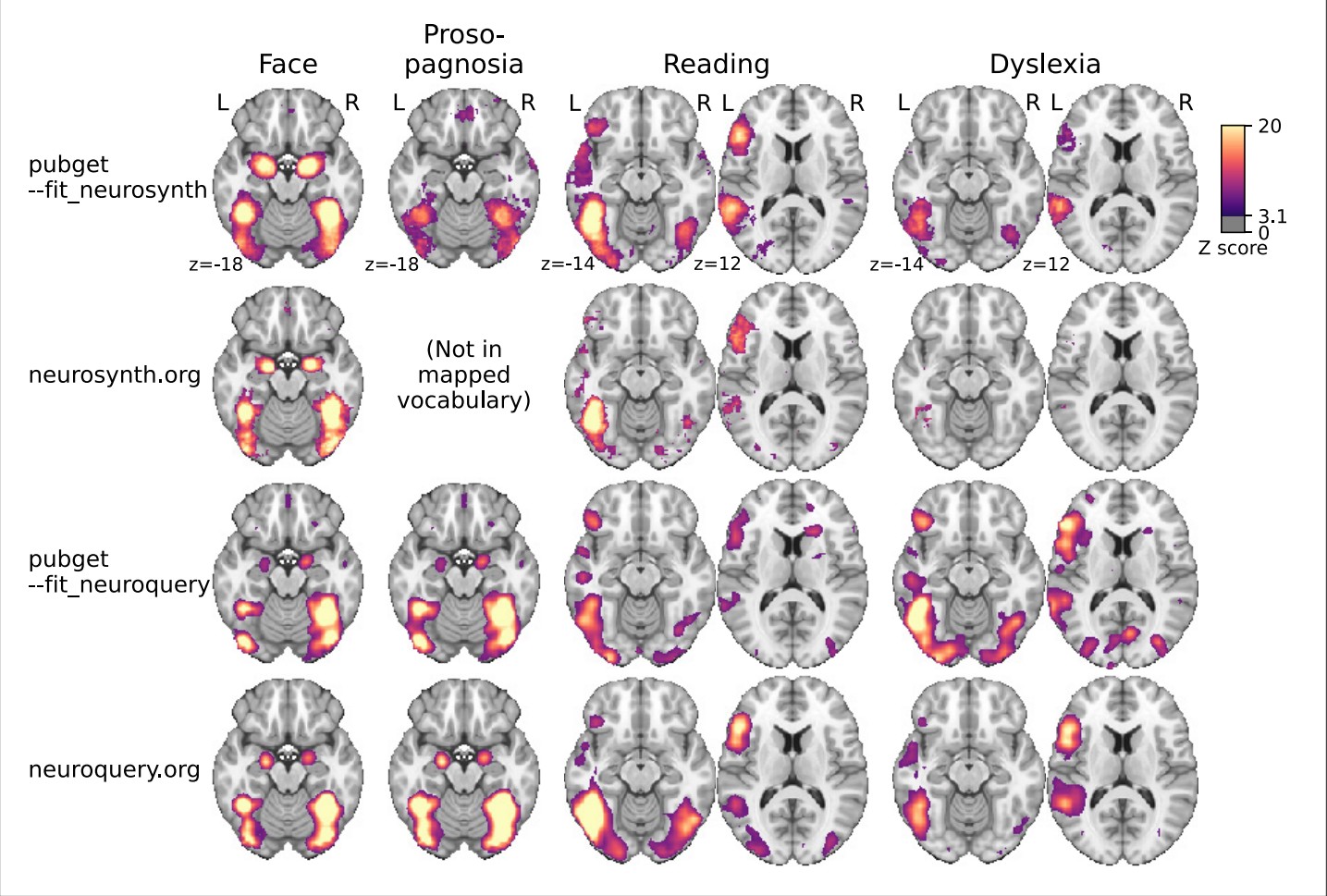

**Figure 3.** Meta-analytic maps produced by pubget and by the original NeuroSynth and NeuroQuery platforms for some example terms. We note that pubget-NeuroSynth (i.e., the top row) has higher statistical power and better face validity for rare terms than the original NeuroSynth (second row). On the other hand, the original NeuroQuery (fourth row) was trained on 13,000 full-text articles and therefore performs better than pubget-NeuroQuery (third row). From these and other examples, we suggest the following rule of thumb: (i) for frequent, well-defined terms such as 'auditory' or 'parietal', all methods produce adequate results; (ii) for formal meta-analysis of a single term, pubget-NeuroSynth produces the best results; (iii) for multi-term queries, neuroquery.org or Text2Brain (*Ngo et al., 2022*) produce the best results. There is no neurosynth.org map for 'prosopagnosia' because this term is too rare to be included in NeuroSynth vocabulary.

extending previous important meta-research papers, as well as for asking new questions that were previously challenging to address. We describe not only neuroimaging but also meta-science examples to demonstrate the benefit of the proposed ecosystem.

These tools are already being used by a large funded project, Neurosynth-compose (*Kent et al., 2024*), an online platform accessible at https://compose.neurosynth.org/ for performing PRISMA-compliant (*Page et al., 2021*) meta-analyses in neuroimaging, as well as smaller research projects.

## Results

To demonstrate the use of the *litmining* tools, we chose examples diverse in their nature, requirements, and scale (from a few dozen documents to tens of thousands).

## Large-scale meta-analyses

For the first time, pubget enables running a complete large-scale meta-analysis as performed by NeuroSynth and NeuroQuery in a fully automated and reproducible manner.

*Figure 3* presents example results obtained running a single pubget command, which performs the full pipeline from data collection to the final statistical analysis and interactive exploration of the resulting brain maps.

Individual neuroimaging studies often lack statistical power and are likely to report spurious findings (*Poldrack et al., 2017*). More reliable associations between brain structures and behavior can be extracted through meta-analysis (*Wager et al., 2007*) by aggregating the *stereotactic coordinates* (coordinates in a standard spatial referential for the brain) reported in these individual studies and performing a statistical analysis to identify consistent brain–behavior associations.

NeuroSynth (*Yarkoni et al., 2011*) made it possible to run such meta-analyses at a large scale by automating the collection of articles and coordinate extraction. NeuroQuery *Dockès et al., 2020* followed a similar approach but introduced a new article collection, an improved coordinate-extraction method, and a different statistical model. Importantly, these projects each share a static corpus, the result of a data collection that has been performed once, but do not provide an easy and reliable way to run the data collection. NeuroSynth does provide some code, but it is slow and difficult to use, and in practice users prefer to rely on the provided static corpus. These corpora are not updated with new publications, and they only contain a subset of the literature, so the practice of reusing these corpora is not sustainable.

Pubget downloads full-text articles from PubmedCentral (PMC) in a fast and reliable way (see section 'Pubget') and extracts existing brain coordinates using the method that generated the Neuro-Query coordinates. This makes it possible to extract coordinates from new sets of publications, enabling large-scale meta-analysis applications and development to move forward without being tied to the static NeuroQuery or NeuroSynth corpora. Moreover, pubget can run large-scale meta-analyses on the data it downloads either by performing the NeuroSynth statistical test or by fitting the Neuro-Query model. The results can be explored through an interactive (local) web application.

We first ran pubget with a query covering many neuroimaging journals as well as the terms "fMRI" (functional magnetic resonance imaging) and "VBM" (voxel-based morphometry). The full query can be found in the https://github.com/neuroquery/pubget/blob/main/docs/example_queries/journal_list_fmri_vbm.txt. This resulted in over 10,000 articles with x, y, z stereotactic coordinates. Although that is a smaller number of publications than the NeuroSynth corpus, which contains over 14,000 abstracts, it represents a much larger amount of text because it also contains the articles full text. Therefore, any given term of interest such as "language" or "aphasia" is seen in more publications, associated with more coordinates, resulting in meta-analyses with higher statistical power. We used pubget to run meta-analyses on this corpus, using the `--fit_neurosynth` and `--fit_neuroquery` options provided by pubget. Some examples are shown in *Figure 3*.

## Participants demographics

This second example reproduces and extends the investigation of sample size and statistical power presented in *Poldrack et al., 2017*. In the original paper, the authors point out that small sample sizes pose a threat to neuroimaging studies' replicability and reliability. *Poldrack et al., 2017* investigate the evolution of the median sample size in the literature through time, exploiting two sources of data: (i) sample sizes for 583 studies manually extracted from published meta-analyses provided by *David et al., 2013* and (ii) new semi-automatic annotations of 548 abstracts in the NeuroSynth corpus. The authors find that the median sample size increased steadily from 1995 to 2015, as well as the number of studies with more than 100 participants.

We used the *litmining* ecosystem tools pubget and labelbuddy to update this study with data from the most recent literature. We further characterized the articles' cohorts and extracted, where possible, participants' age, sex, and whether participants were patient or healthy control. Using pubget, we downloaded papers from PubMed Central that mention the term "fMRI" in their abstracts. This yielded 4230 articles.

We then used labelbuddy to explore and annotate a first set of 20 papers and implement a plugin as a simple set of rules to automatically extract the sample size and participants demographics from the text, shared on GitHub. With labelbuddy we annotated a new batch of 100 articles and obtained a 10% mean absolute percentage error for the automatic extraction. As this is satisfactory for this illustration, we then ran the extraction on the full set of 4230 articles to obtain the results presented below.

To illustrate how our ecosystem facilitates the evaluation of newly emerging language models, we also extracted sample sizes using zero-shot learning using Generative Pre-trained Transformer 3.5 (GPT-3.5), a commercially available large language model. We first prototyped chat-completion prompts to extract sample size from a scientific text sample as well as pre- and post-processing pipeline using a set of 188 manually annotated papers. We then applied the GPT pipeline to an unseen set of 103 annotated papers to evaluate model performance.

We found that GPT-3.5 performed better than our heuristic approach, with the heuristic finding the exact sample size in 36% of papers, compared to 50% for GPT-3.5. The two approaches had similar error rates when they did make a guess; the heuristic had a median absolute percent error of 0% (median absolute error of 0 participants), and GPT-3.5 had a median absolute percent error of 0.9% (1 participant). However, GPT-3.5 had better recall; it made a guess for 100% of evaluation papers, whereas pubextract made a guess in 54%. This is likely due to the fact that GPT-3.5 is able to extract sample sizes from more complex sentences than our heuristic, which is limited to simple sentences. The overall median sample size in the validation set was 32 participants, and the median sample sizes extracted by pubextract and GPT-3.5 were 24 and 34, respectively. *Figure 4A* shows the performance of both methods. Note that we identified a bug in GPT-3.5 embedding search implementation after the evaluation was completed; these present results reflect the revised implementation.

The purpose of this illustration is not to maximize the performance of these methods, but to show that their comparison was extremely easy using our tools; this example of model comparison was completed in a couple of weeks. While the accuracy of sample size extractors could be improved, we replicated and extended the results presented in *Poldrack et al., 2017*, and we found them sufficiently accurate for estimating the median sample size in the literature.

In *Figure 4C*, we show the median sample size evolution's through time according to the four available sources: data from *David et al., 2013*, semi-automatic annotations of NeuroSynth abstracts from *Poldrack et al., 2017*, as well as automatic extractions from papers collected with pubget in the present study, using pubextract and GPT-3. We note that the median sample size estimated from the new data seems slightly smaller in 2015 than what was extracted from NeuroSynth abstracts. This may be due to errors in the automatic extraction, as well as the fact that papers with large sample sizes are more likely to mention it in the abstract.

We also investigated the distribution of participant ages reported in articles. We automatically extracted the age range, the mean age, and the age standard deviation when they were reported. In *Figure 4B*, we plot the distribution of the mean age reported in individual studies. We note that healthy participants very often have the typical age of university students, possibly reflecting the declaimed practice of some studies that recruit students from their own department as participants (*Henrich et al., 2010*).

As high-quality annotations about participants are useful for many projects (e.g., as validation data for efforts to extract Population, Intervention, Comparison, Outcome data with machine learning), our annotations are shared online, version controlled, and may be expanded in the future. The annotation https://litmining.github.io/labelbuddy-annotations/projects/participant_demographics.html provides rich information about the (often complex) group structure of participants and demographic information. Annotating this information is made easy by labelbuddy's interface and by a utility tool provided by *litmining* located in the annotations repository that infers the participant group structure and information about each subgroup and displays it as annotations are being added (*Figure 5*).

At the time of writing this annotation, repository contains over 3K annotations for 188 documents.

## Dynamic functional connectivity methods

This third example illustrates the use of *litmining* on a fully manual project that relies only on pubget and labelbuddy without adding any custom code beyond simple plotting.

In recent years, dynamic functional connectivity (dFC) has emerged as a popular approach for characterizing the time-varying patterns of functional connectivity in the brain. Numerous methods have been proposed for assessing dFC, but the selection of a specific method for a given study is often arbitrary and lacks a clear rationale. Additionally, despite being frequently cited in the literature, some dFC assessment methods are seldom applied in practice.

In this study, we investigated six different dFC assessment methods and their usage in the literature. We downloaded 385 papers from PubMed Central using pubget and the search query "(dynamic

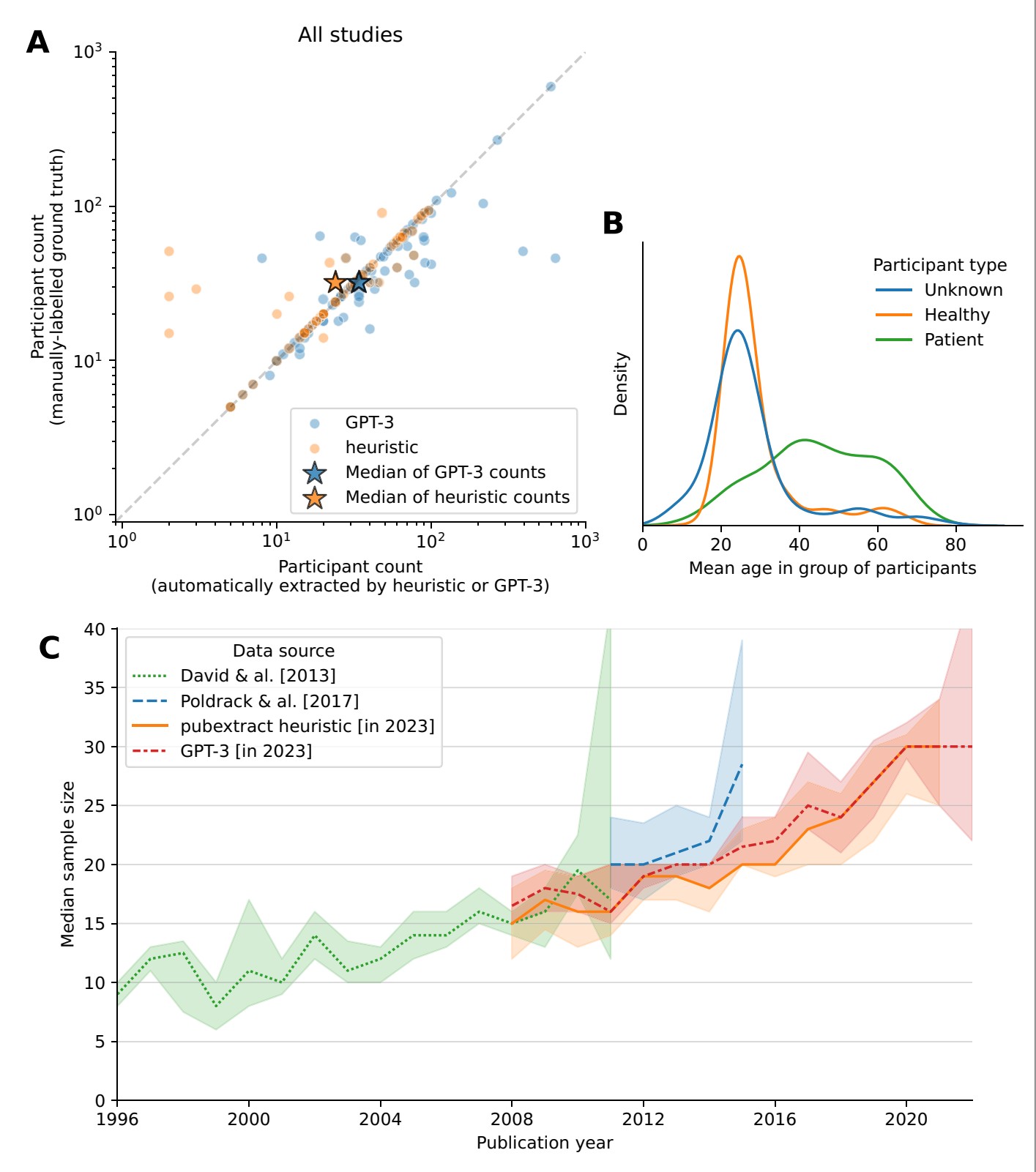

**Figure 4.** Extraction of participant count and demographics from articles' text. (**A**) Performance of the sample size extraction plugin in pubextract as well as GPT-3.5. The x-axis shows the sample size extracted from the text, and the y-axis shows the sample size reported in the article. The dashed line shows the identity line. The stars represent the median values of each extraction method compared to the ground-truth median. (**B**) Distribution of ages for different participant groups, extracted with pubextract. 'Unknown' is chosen when the tool fails to detect whether a participant group corresponds

*Figure 4 continued on next page*

Dockès, Oudyk *et al.* eLife 2024;13:RP94909. DOI: https://doi.org/10.7554/eLife.94909                                    8 of 19

*Figure 4 continued*

to patients or healthy controls – in most cases when this is not specified explicitly, the participants are healthy. We note that the distribution of healthy participants' ages has a large peak around the age of university students, who are often recruited in their own university's studies. Patients tend to be older on average, with a long tail likely due to studies on aging or neurodegenerative diseases. (**C**) Median sample size through time. Error bars show 95% bootstrap confidence intervals. Following *Poldrack et al., 2017*, for sample sizes extracted from pubget-downloaded articles, we only consider single-group studies.

functional connectivity[Abstract]) OR time-varying functional connectivity[Abstract]". These papers cover a range of topics, including clinical, cognitive, and methodological studies.

We then used labelbuddy to manually annotate a subset of papers with the dFC assessment method(s) used in the study and the application of the study (e.g., clinical, cognitive). This information provides valuable insights into the current state of the field as it helps identify which dFC assessment methods are most commonly used and in which types of studies. The annotations are available online.

The scatter plot in *Figure 6* presents the number of times each method was applied across the annotated papers versus the number of citations received. A total of 70 papers were annotated for this analysis. This figure serves to examine the relationship between the practical adoption of each method and its citation count. Essentially, it explores whether a method that receives frequent citations is also commonly employed in practice.

## Discussion

We have introduced an ecosystem of simple tools that help make text mining of the biomedical literature more accessible, transparent, and reproducible. This ecosystem, when used with recommended platforms like GitHub, fosters collaborative, reproducible research. The ecosystem and recommendations include:

(i) **Pubget**, a tool for downloading and processing articles from PMC.
(iii) **Labelbuddy**, a flexible and effective text labeling application.

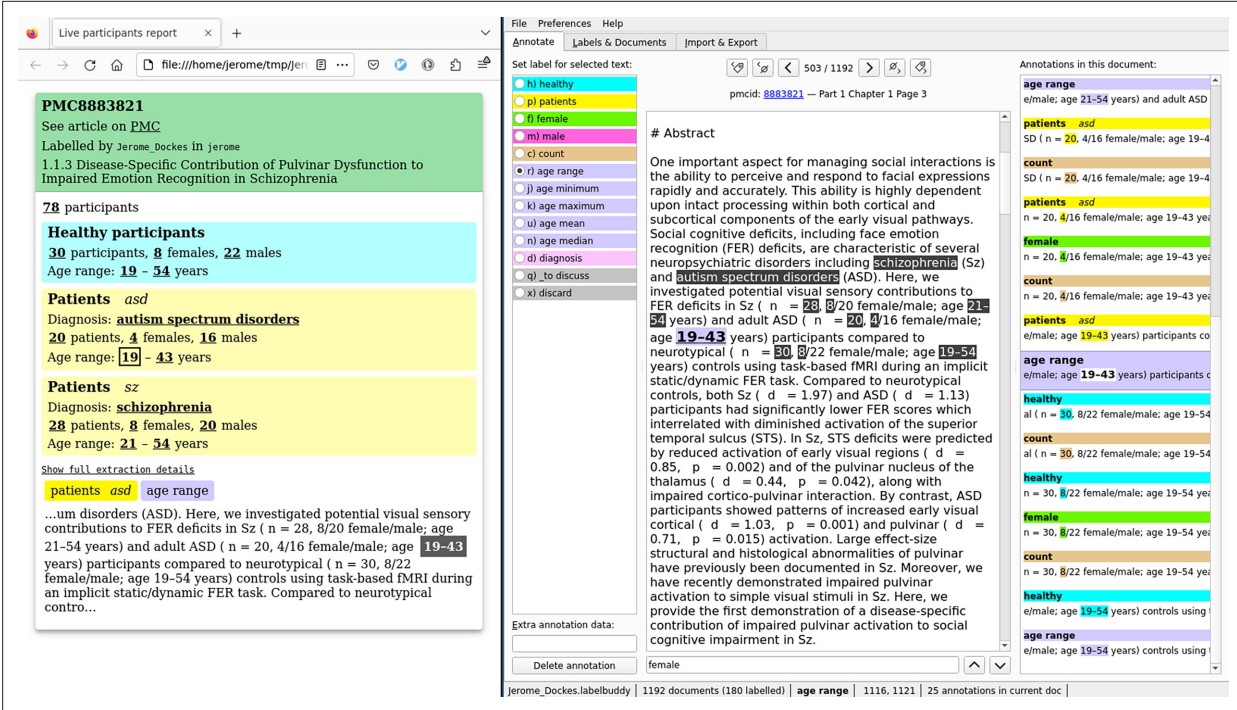

**Figure 5.** Screen capture of annotating the participants' demographic in one document. The window on the right is labelbuddy, displaying an article, the available labels and their shortcuts (right column), and the existing annotations for the current document (right column). The left window is the dedicated tool for participant demographics, showing the inferred group structure and information about each group. This tool is not part of labelbuddy itself; it is distributed as part of the labelbuddy-annotations repository.

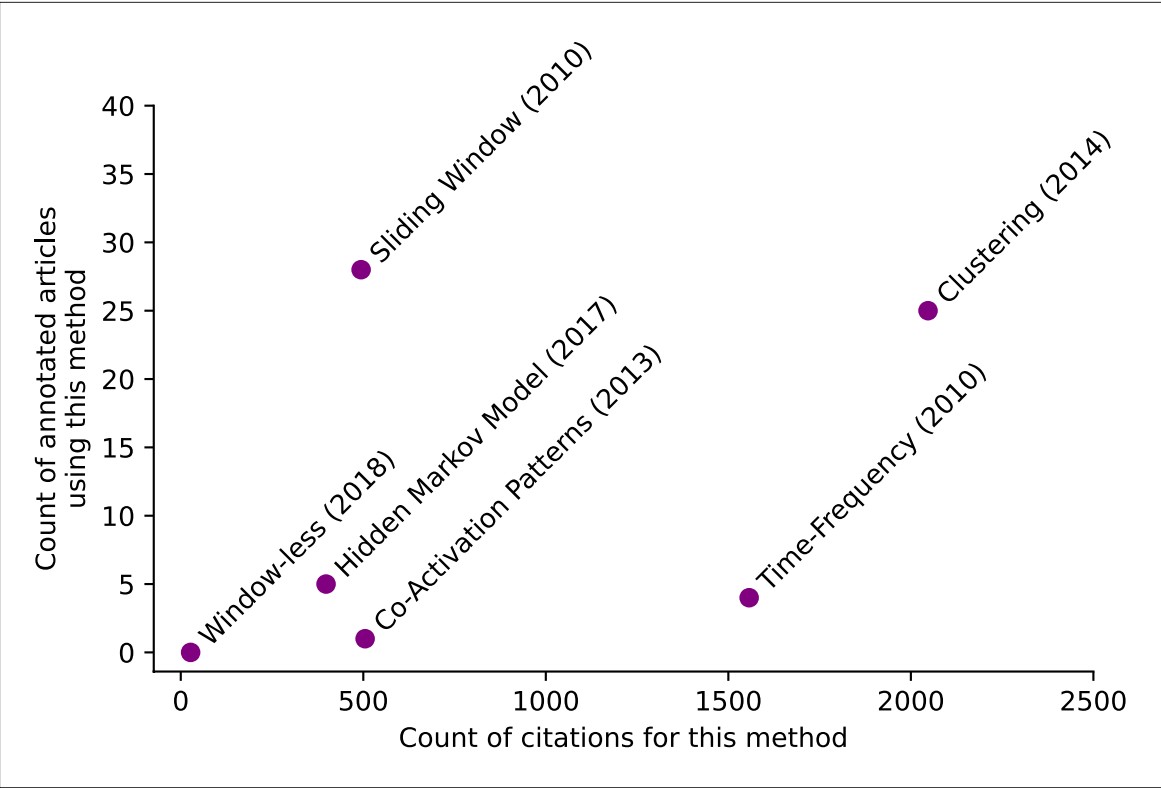

**Figure 6.** The scatter plot displays the number of times each dynamic functional connectivity (dFC) assessment method was applied across the annotated papers versus the number of times it was cited. The year of the first publication of each method is also shown besides its name. *Clustering* was both highly cited and highly applied, while *Time-Frequency* was highly cited but not highly applied. *Sliding Window* was highly applied, although not as frequently cited as *Clustering*. *Co-Activation Patterns* and *Window-less* were not highly applied, although *Co-Activation Patterns* was as frequently cited as *Sliding Window*.

(iii) **Labelbuddy-annotations**, a repository of manual annotations of the neuroimaging literature made with labelbuddy.

(iv) **Pubextract**, a Python package to extract information from the output of pubget.

These tools allowed us to run for the first time fully automated, large-scale meta-analyses whose provenance can be tracked. Further, we were able to reproduce, update, and automate a systematic analysis of sample sizes in fMRI studies and further demonstrated the usefulness of our ecosystem for model comparison, which is an essential component of literature mining. Finally, we could efficiently review dFC methods use in a systematic and transparent way. These examples illustrate the potential of the proposed ecosystem to facilitate meta-research and literature mining.

There are wide-ranging applications of these tools, from reviews and meta-analyses that summarize the literature, to meta-research evaluating and improving research itself, to the creation of new corpora for text mining, as well as evaluation of different natural language processing models. These tools can be used to generate new corpora as well as check the performance of existing tools that work with text data. The accessibility, scalability, and reproducibility of these tools are essential for updating past studies; as the literature grows with new information and evolves with new methods and practices, it is critical to make updates of reviews, meta-analyses, and other meta-research as seamless as possible. Our tools will make it easier to evaluate new models as they are developed, opening the doors of literature mining to researchers with a wider range of technical expertise and domain knowledge. Because pubget gathers biomedical literature provided by PubMed Central, these tools may be used in any biomedical field. Further, labelbuddy has even greater flexibility; it could be used to annotate any kind of text data, not just research articles.

While this ecosystem constitutes an important first step and facilitates a wide variety of projects, it has several limitations. A limitation of pubget is that it is restricted to the Open-Access subset of PMC, and this excludes a large portion of the biomedical literature. This choice was made to make pubget

maintainable and is unlikely to be revisited in the near future. Because it relies on a single, stable, and open resource, pubget is less likely to be affected by changes in the services it depends on, to fail for some users due to authentication issues, or to face high maintenance costs due to the need to handle multiple, potentially evolving formats chosen by individual publishers. It also makes it possible to regularly test pubget without needing an account for a paid resource. Finally, it allows users to conduct their analyses without worrying about infringing complex copyright laws as all articles in the Open Access subset have Creative Commons or similar licenses and are available for text-mining purposes according to the PMC online documentation. While the restriction to PMC Open Access is an important limitation for reviews or manual meta-analyses that aim to be exhaustive, it is a smaller problem for large-scale meta-analysis or meta-science projects that only need a large, representative sample of the literature. Moreover, when the Open Access is insufficient, pubget can still be useful by automatically handling a large part of the articles, leaving the researchers with fewer articles to collect from other sources and process. Finally, there is hope that PMC will cover an even larger portion of the literature in the future as there is a trend toward more open-access papers (e.g., *White House, 2022*). Another potential issue, as with all recent tools that do not yet have a mature community of users and contributors, is the perennity of pubget, labelbuddy, and of the annotations repository. We have alleviated this risk through design choices and the distributed nature of these tools. Indeed, pubget and labelbuddy are local applications that run on the users' machines, lifting the high maintenance costs of a web service. Moreover, they both rely on a small set of widely used and well-maintained libraries, and intentionally have a restricted scope and feature set, to reduce the maintenance cost. Perhaps more importantly, they store their output in standard formats (CSV and JSON) in files or directories chosen by the user on their local disk and in Git repositories. This makes it easy to use the data without the tool, or to switch to a different tool. Even labelbuddy's internal, binary format is the universally supported SQLite file format. By contrast, when relying on tools that store data online or even in local databases, or those that use complex or proprietary file formats, the user needs to worry about losing their work or spending significant time extracting and reformatting it if they decide to stop using the application. Similarly, due to the distributed nature of Git, annotations in labelbuddy-annotations repository will not be lost if the original authors of the repository stop contributing or even if the online repository is deleted – all contributors have a local copy and a new online repository can easily be created on several platforms such as GitHub, GitLab, or BitBucket. Finally, although they are interoperable, the *litmining* tools are completely independent of each other (beyond pubget's option to store result in labelbuddy's JSON format) and from the annotations repository, and they are only linked by project development guidelines. We expect that many users will opt to use only part of the ecosystem.

One downside of this distributed and local design is a less-integrated experience for users. Compared to a solution where a researcher would perform all tasks on one online platform, they have to take the extra steps of installing the tools (which is seamless on Linux, MacOS, and Windows) and managing their local files. In exchange, researchers gain full control of the work they produce and flexibility in how they store, analyze, and share their work. Moreover, the tools we introduce can be used as building blocks for more integrated and comprehensive solutions for specific tasks. For example, the Neurosynth-compose (*Kent et al., 2024*) online platform is a complete solution for conducting reproducible neuroimaging (manual) meta-analyses (accessible at https://compose. neurosynth.org/). Neurosynth-compose goes well beyond what the litmining ecosystem can offer for conducting manual meta-analyses. However, the development of the (much larger) Neurosynth-compose project still benefits from the litmining tools: pubget is used for part of the data collection, and both pubget and labelbuddy are used for prototyping and validating (through manual annotations) some upcoming features of the platform, such as information extraction with neural networks.

The ecosystem can be used from the command line or Python (e.g., when using pubget to download articles). However, we have made an effort to make these tools as accessible as possible, with minimal need for programming, while maintaining excellent reproducibility and scalability. To mitigate this challenge, we have created step-by-step instructional videos on how to use our ecosystem to perform an entire literature-mining project. These can be found at our central website, https://litmining.github.io/.

Overall, despite these limitations and the intentionally restricted scope of the tools introduced, the proposed ecosystem constitutes an important step toward a more accessible open, reproducible and collaborative biomedical – and in particular neuroimaging – meta-science.

## Materials and methods

### Pubget

Pubget is a command-line tool for downloading and processing publications from PMC. It is written in Python. This section provides a brief overview, and more information on using pubget can be found in the https://neuroquery.github.io/pubget/pubget.html.

The user provides either a PMC query, such as "fMRI[Abstract]", or a list of PMC IDs. Pubget downloads all the corresponding publications that are in PMC's Open Access subset and processes them to extract their content. Outputs are stored in CSV files, making them very easy to use for subsequent analysis steps with scientific software such as Python, R, or MATLAB.

A pubget execution proceeds in several steps, detailed below. The first steps consist of downloading the articles and extracting their content, and these are always executed. Then, several optional steps can be added that prepare the downloaded data for more specific uses – for example, extracting Term Frequency–Inverse Document Frequency (TF-IDF) features for use in natural language processing, or preparing documents in labelbuddy's format for easy annotation.

Each step stores its output in a separate directory. When invoking pubget again with the same query, steps that have already been completed are not re-run. Most steps can process different articles in parallel.

### Downloading articles

The first step is to download the articles matching a PMC query. This is done with the https://www.ncbi.nlm.nih.gov/books/NBK25497/. This collection of web services provides a programmatic interface to the National Library of Medicine (NLM) resources. Pubget strives to provide a client that is reliable and robust to network and server failures. As an example, the NLM servers often send responses with a 200 (success) status code, despite the fact that an internal server error has occurred and the response content is an error message. Pubget handles these cases by checking the content of the responses and retrying the download when necessary.

Pubget also respects the request rate limits mentioned in the E-Utilities' documentation to avoid overloading these resources. pubget includes in requests the API key when provided by the user, which is not required but encouraged by the NLM.

Given a query, pubget first uses E-Search to query the PMC database and build a set of matching PMCIDs. The result is stored on the Entrez history server, and pubget stores the keys that identify the result set on the history server. Then, pubget uses E-Fetch to download the XML documents, in batches of 500 articles each. Failed downloads are attempted several times.

If pubget is executed again with a query for which a partial download exists, only missing batches of articles are downloaded. This is possible due to the Entrez history server keys stored by pubget, which identify a fixed set of articles.

### Extracting the articles' content

The next step is extracting content in a usable format from the XML batches. Pubget first extracts articles from the downloaded batches and stores them in separate files. Then, each article's XML is parsed and useful information is extracted. This is done in parallel across articles using a number of processes chosen by the user. Metadata such as authors or publication year are extracted with XPath. The articles are transformed with an XSLT stylesheet to obtain the text, free of XML markup.

Tables require special attention. The XML language used by PMC, JATS, allows two different table formats – XHTML or OASIS. Pubget relies on the DocBook stylesheets to transform articles to the XHTML format and extracts and processes the tables from there. Tables are stored in pubget's output in their original XML format, in the XHTML format, and as CSV files.

Stereotactic coordinates are extracted from tables with the approach that was used to create the neuroquery corpus (*Dockès et al., 2020*). The coordinate space (MNI or Talairach) is detected with the heuristic used by NeuroSynth (*Yarkoni et al., 2011*).

## Meta-analyses

Pubget can optionally extract TF-IDF features from articles and run the same chi-square test as NeuroSynth, or train a neuroquery model. A small Python script is added to the output directory for a NeuroSynth or a NeuroQuery meta-analysis, which, when executed, allows a user to explore the meta-analytic maps interactively, with an interface similar to that of https://neurosynth.org/.

## Integration with other tools

Pubget aims to make it as easy as possible to use the data it generates for subsequent analysis steps. Part of this effort consists in formatting this data to target specific tools that are used by the community or well-suited for common analysis steps. At the time of writing, such integrations exist only for two external tools (labelbuddy and NiMARE), but we are planning to add more, depending on user feedback.

Labelbuddy is a desktop application for manually annotating documents, described in more detail below. Pubget can format its output in the JSONLines format used by labelbuddy. Therefore, the pubget output can be directly imported in a labelbuddy database and the user can start labeling these documents without any effort.

NiMARE (*Salo and Bottenhorn, 2018*) is the established Python library for neuroimaging meta-analysis. It provides implementations of most of the widely used meta-analysis methods. Pubget can format its output in the JSON format used by NiMARE to represent coordinate-based neuroimaging datasets and run meta-analyses. As a result, pubget output can directly be loaded with NiMARE and processed by any meta-analysis nimare method. For rigorous manual meta-analysis, additional steps to manually filter articles, select contrasts and check the extracted coordinates will need to be applied. However, having the pubget output in NiMARE format both facilitates this task and enables obtaining quickly a crude approximation of the final meta-analytic maps.

## Plugins

A core design element of pubget is that it easily allows defining plugins. This means that additional processing steps can be developed and distributed independently of pubget and still become part of the pubget command when they are installed. Such plugins are discovered at runtime and provide additional commands or options to the pubget command-line interface. One possible use for this plugin mechanism is to distribute processing steps that depend on libraries that are not installed by pubget, or that are too specific to be directly integrated in the main tool. These plugins foster reuse and an opportunity for projects to collaborate and progress at a faster pace or with less strict coding standards than pubget itself. A collection of such plugins is provided in the https://github.com/neuro-datascience/pubextract package. Among other plugins, it adds to pubget the `--participants` option, which extracts participant counts and demographic information. This plugin is described in more detail below, where we present the methods used for the participant demographics example.

## Labelbuddy

Although many annotation tools exist, we did not find a good solution when we searched for one adapted to our needs for extracting information from the neuroimaging literature. Most available tools are web-based and annotators collaboratively contribute to a centralized database of annotations. This is necessary when many annotators are involved, which is typically the case when labeling text for machine learning, as many annotations are then needed. However, this makes the setup and integration in a Git-based workflow more difficult. As a result, many projects prefer to use a simple spreadsheet at the cost of relying on a much less convenient interface and losing the provenance and location of the exact portions of text that support the conclusions.

For small research projects with a few annotators, labelbuddy can compete both with web-based annotation tools running on a local server and with more ad hoc local tools such as text editors or spreadsheets. It has an intentionally minimal feature set and focuses only on creating the annotations – for example, it does not create reports, compare annotators, etc. It can seamlessly be installed on Linux, Windows, or MacOS and does not require any setup or configuration. Labelbuddy is also distributed as standalone binaries for those three platforms, so it can be used by downloading and running a single executable without any specific installation.

**Table 1.** Number of documents, labels, annotators, and annotations for each project in the labelbuddy-annotations repository.

| Project name | Documents | Labels | Annotators | Annotations |
|---|---|---|---|---|
| review-neuro-meta-analyses | 421 | 26 | 1 | 912 |
| cluster_inference | 193 | 20 | 2 | 1610 |
| participant_demographics | 188 | 15 | 10 | 3006 |
| dynamic_functional_connectivity | 70 | 9 | 1 | 94 |
| parkinsons | 60 | 6 | 1 | 411 |
| NER_biomedical | 11 | 9 | 1 | 58 |
| neurosynth_use | 8 | 4 | 1 | 20 |
| autism_mri | 5 | 21 | 1 | 69 |
| neurobridge_fmri | 0 | 0 | 0 | 0 |
| Total | 930 | 102 | 14 | 6180 |

Internally the documents, labels, and annotations are stored in a single ordinary file. They can be exported to JSON, which is easy to parse in any programming language. Any analysis or further manipulation of the annotations can be done with the researchers' preferred tool, such as Python or R. Aggregating annotations across annotators and tracking their evolution can be done with the preferred tool for collaboration and version control – Git and a remote repository. Although it was designed with small projects in mind, labelbuddy can easily handle dozens of thousands of documents, large documents, and hundreds of thousands of annotations in a single file even on a small machine, making it a lightweight tool.

Pubget is well integrated with labelbuddy and can store the extracted text in JSON files that can be directly imported into labelbuddy for annotation, providing a core element of the *litmining* ecosystem.

## Labelbuddy-annotations: A Git repository of manual annotations

To store and showcase annotations from several projects, we created the https://litmining.github.io/labelbuddy-annotations/overview.html Git repository. It stores documents, labels, and annotations from several projects and annotators. The documents have typically been downloaded with pubget, although that is not a requirement. To start adding annotations, contributors simply need to create annotations in labelbuddy, export them, commit the resulting JSON, and open a Pull Request. A Jupyter book showcases the contents of the repository, provides documentation for users and contributors, and presents simple analyses of the available data. It is hosted on GitHub Pages and updated by Continuous Integration (GitHub Actions). Moreover, the repository also contains a small Python package to facilitate the analysis of existing annotations. *Table 1* shows a summary of the projects currently included in the repository.

## Participant demographics

To extend the study of sample sizes presented in *Poldrack et al., 2017*, we used two approaches to extract the number of participants from the text of a study: a manually defined heuristic and a workflow using OpenAI's pre-trained GPT large language model.

## Pubextract heuristic

The heuristic approach proceeds in two steps. The first step recognizes mentions of participant groups in the text and extracts their different components (such as count and mean age). This is done with simple string matching and hard-coded rules – it does not involve any statistical learning. These rules are described by an Extended Backus–Naur Form grammar and parsing is performed by the https://lark-parser.readthedocs.io/en/latest/index.html Python library. The second step aggregates the different group mentions detected in the first step and infers the participant group structure. The full implementation can be found on GitHub as part of the https://github.com/litmining/pubextract.

## GPT workflow

The GPT approach consists of two steps: a semantic search to identify the most relevant section of each paper and a zero-shot prompt to extract participant count from a given section. First, we separated each paper into chunks of less than 2000 characters using Markdown headers from the Body of each paper. If after exhausting all headers a section was above 2000 characters, we further subdivided it using newline characters, to reach a chunk under the desired length. Next, using the OpenAI API, we generated a 1536-dimensional embedding for all chunks in each article using the pre-trained 'text-embedding-ada-002' model. To find the chunk most likely to contain participant demographic information for each article, we embedded a string search query using the same embedding model and computed the Euclidean distance between each chunk and the search embedding. After piloted various search queries, we settled on the following search query: 'How many participants or subjects were recruited for this study?'. Finally, for each paper we attempted to extract participant demographics using a zero-shot prompt. Using OpenAI Chat Completion API, we prompted "GPT-3.5-turbo" using a template that inserted text from the nearest chunk from each article, and the instruction to "identify groups of participants that participated in the study, and underwent MRI." To ensure that GPT provided a structured response, we included a JSON Schema for the desired output to the model as part of the prompt, using the "function calling" parameter. If the model responded with a null object for a chunk, we iteratively reattempted extraction on the next-nearest chunk until a response was generated. Finally, we applied a manually derived set of cleaning results to the final output of the model, such as dropping rows with missing values or predicted counts of "0". The full implementation can be found on GitHub as part of the publang package.

## Validation

To validate the sample size extraction methods, we manually annotated 291 papers, which we divided into a training set of 188 and a final validation set of 103 papers. We used the training example to iteratively improve the heuristic extraction rules, and the search query and extraction prompt used in the GPT approach (i.e., "prompt engineering").

The code for performing this validation and the plots shown in this paper, as well as the manual annotations, can be found on GitHub. As detailed annotations about study participants' demographics are useful on their own and for other projects, the annotation effort continues in the https://litmining.github.io/labelbuddy-annotations/projects/participant_demographics.html repository.

To validate the age extraction, we opened the automatically generated annotations in labelbuddy and checked them manually. In these papers, we found no errors in the age extraction.

Automatically extracting information about study participants from text is a difficult task and beyond the scope of the current paper. Our heuristic approach leaves much room for improvement, and statistical methods, for example based on pre-trained deep language models, would probably yield better results. What we aim to illustrate with this example project is that by using pubget we were able to focus on the task of interest (finding and analyzing the participants' information), without spending time and effort on obtaining the relevant text. Moreover, the ability to easily annotate the documents with labelbuddy, and to visualize automatically generated annotations with the same tool, was key to an efficient validation step.

## Data and code availability

### Our tools

- Pubget: The source code for pubget is available on GitHub under an MIT License at https://github.com/jeromedockes/pubget.
- Labelbuddy: The source code for labelbuddy is available on GitHub under a GNU General Public Licesnce (version 3) at https://github.com/jeromedockes/labelbuddy/.
- Labelbuddy-annotations: The repository containing the manual annotations and code to work with the annotations is available on GitHub under a MIT license at https://github.com/jeromedockes/labelbuddy/ . The exact annotations used at the time of the writing of this paper is available statically on Zenodo under an MIT License at https://zenodo.org/records/15225229.
- Pubextract: The package used to extract information from the output of pubget is available on GitHub under an MIT License at

### Application: Large-scale meta-analyses

- Papers: We used a collection of fMRI papers downloaded with pubget from the https://pmc.ncbi.nlm.nih.gov/tools/openftlist/; *National Library of Medicine, 2003* shared under various https://pmc.ncbi.nlm.nih.gov/tools/textmining/. The list of PMCIDs included in this collection can be found on GitHub.
- Code: The code used to create *Figure 3* is available on GitHub under an MIT License at https://github.com/neurodatascience/literature_mining_paper/blob/main/figure_scripts/compare_meta_analyses.py (copy archived at *Dockès, 2025a*).

### Application: Participant demographics

- Papers: We used a collection of fMRI papers downloaded with pubget from the https://pmc.ncbi.nlm.nih.gov/tools/openftlist/; *National Library of Medicine, 2003* shared under various https://pmc.ncbi.nlm.nih.gov/tools/textmining/. The list of PMCIDs included in this collection can be found on GitHub.
- Annotations: The set of manual annotations used for the validation of the sample size extraction is available on GitHub under an MIT License in the https://github.com/litmining/labelbuddy-annotations/tree/main/projects/participant_demographics repo. The exact annotations used at the time of the writing of this paper are available statically on Zenodo under an MIT License at https://zenodo.org/records/15225229.
- Software: The package used for information retrieval using large language models (here, GPT-3.5) is available on GitHub under a BSD 3-Clause License at https://github.com/adelavega/publang (copy archived at *de la Vega, 2025*)
- Custom code: The code used for the validation of the sample size extraction methods and to create *Figure 4* is available on GitHub under an MIT License at https://github.com/jeromedockes/fmri_participant_demographics (copy archived at *Dockès, 2025b*).
- Annotation guide: An annotation guide for this project is available on the https://litmining.github.io/labelbuddy-annotations/projects/participant_demographics.html.

### Application: Dynamic functional connectivity

- Papers: We used a collection of dFC papers downloaded with pubget from the https://pmc.ncbi.nlm.nih.gov/tools/openftlist/; *National Library of Medicine, 2003* shared under various https://pmc.ncbi.nlm.nih.gov/tools/textmining/. The list of PMCIDs included in this collection can be found on Github.
- Annotations: The set of manual annotations used to create *Figure 6* is available on GitHub under an MIT License in the https://github.com/litmining/labelbuddy-annotations/tree/main/projects/dynamic_functional_connectivity repo. The exact annotations used at the time of the writing of this paper are available statically on Zenodo under an MIT License at https://zenodo.org/records/15225229.
- Custom code: The code used to create *Figure 6* is available on GitHub under an MIT License at https://github.com/neurodatascience/literature_mining_paper/blob/main/figure_scripts/dFC_plot.py (copy archived at *Dockès, 2025a*).

## Acknowledgements

This work was partially funded by the Tanenbaum Open Science Institute at The Neuro, the National Institutes of Health (NIH) NIH-NIBIB P41 EB019936 (ReproNim) The Canadian Institutes of Health Research (CIHR PJT-185948, PJT-197805), the Fonds de Recherche du Quebec, The Natural Sciences and Engineering Research Council of Canada (RGPIN/03543-2021), the National Institute of Mental Health (R01MH096906, Neurosynth), the Michael J. Fox Foundation, the Quebec Parkinson Network, the McConnell Brain Imaging Centre, the Canada First Research Excellence Fund, awarded to McGill University for the Healthy Brains for Healthy Lives initiative (NeuroHub), the Chan Zuckerberg Initiative (EOSS5-0000000401) and the Brain Canada Foundation with support from Health Canada, through the Canada Brain Research Fund in partnership with the Montreal Neurological Institute.

# Additional information

## Funding

| Funder | Grant reference number | Author |
|---|---|---|
| Fonds de Recherche du Québec - Santé | 291772 | Kendra M Oudyk |
| Natural Sciences and Engineering Research Council of Canada | 256648 | Jean-Baptiste Poline |
| National Institute of Mental Health | R01MH096906 | Jean-Baptiste Poline |
| Tanenbaum Open Science Institute at The Neuro | | |
| National Institutes of Health | P41 EB019936 (ReproNim) - funded meetings | |
| Canadian Institutes of Health Research | PJT-185948 | |
| Canadian Institutes of Health Research | PJT-197805 | |
| Fonds de recherche du Québec | | |
| Natural Sciences and Engineering Research Council of Canada | RGPIN/03543-2021 | |
| Michael J. Fox Foundation | | |
| Quebec Parkinson Network | | |
| McConnell Brain Imaging Centre | | |
| Canada First Research Excellence Fund | Healthy Brains for Healthy Lives initiative (NeuroHub) | |
| Chan Zuckerberg Initiative | EOSS5-0000000401 | Mohammad Torabi |
| Brain Canada Foundation | | |

The funders had no role in study design, data collection and interpretation, or the decision to submit the work for publication.

## Author contributions

Jérome Dockès, Conceptualization, Resources, Data curation, Software, Formal analysis, Supervision, Funding acquisition, Validation, Investigation, Visualization, Methodology, Writing – original draft, Project administration, Writing – review and editing; Kendra M Oudyk, Conceptualization, Resources, Data curation, Software, Formal analysis, Funding acquisition, Validation, Investigation, Visualization, Methodology, Writing – original draft, Writing – review and editing; Mohammad Torabi, Conceptualization, Data curation, Formal analysis, Investigation, Visualization, Methodology, Writing – original draft; Alejandro I de la Vega, Conceptualization, Resources, Data curation, Software, Formal analysis, Validation, Investigation, Visualization, Methodology, Writing – original draft, Writing – review and editing; Jean-Baptiste Poline, Conceptualization, Supervision, Funding acquisition, Writing – original draft, Project administration, Writing – review and editing

## Author ORCIDs

Jérome Dockès ⓘ https://orcid.org/0000-0002-5304-2496
Kendra M Oudyk ⓘ https://orcid.org/0000-0003-4087-5402
Mohammad Torabi ⓘ https://orcid.org/0000-0002-4429-8481
Alejandro I de la Vega ⓘ https://orcid.org/0000-0001-9062-3778
Jean-Baptiste Poline ⓘ https://orcid.org/0000-0002-9794-749X

Reviewer #1 (Public Review): https://doi.org/10.7554/eLife.94909.2.sa1
Reviewer #2 (Public Review): https://doi.org/10.7554/eLife.94909.2.sa2
Reviewer #3 (Public Review): https://doi.org/10.7554/eLife.94909.2.sa3
Author response https://doi.org/10.7554/eLife.94909.2.sa4

# Additional files

## Supplementary files
MDAR checklist

## Data availability
Custom code is available at Zenodo. For full details please see Materials and methods, Data and code availability.

The following dataset was generated:

| Author(s) | Year | Dataset title | Dataset URL | Database and Identifier |
| --- | --- | --- | --- | --- |
| Dockès J, Oudyk K, Vega A, Torabi M, Chen A, Gau R, Kennedy D, McPherson B, Wang M, Mirhakimi M, Poline J | 2025 | litmining/labelbuddy-annotations: Time when the data matched our 'Mining the neuroimaging literature' paper | https://doi.org/10.5281/zenodo.15225229 | Zenodo, 10.5281/zenodo.15225229 |

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
