## [Editor Report · eLife assessment]

The study presents an **important** ecosystem designed to support literature mining in biomedical research, showcasing a methodological framework that includes tools like Pubget for article collection and labelbuddy for text annotation. The **solid** evidence presented for these tools suggests they could streamline the analysis and annotation of scientific literature, potentially benefiting research across a range of biomedical disciplines. While the primary focus is on neuroimaging literature, the applicability of these methods and tools might extend further, offering an advance in the practices of meta-research and literature mining.

---

## [Referee Report · Reviewer #1 (Public Review)]

Summary:

In this paper, the authors present new tools to collect and process information from the biomedical literature that could be typically used in a meta-analytic framework. The tools have been specifically developed for the neuroimaging literature. However, many of their functions could be used in other fields. The tools mainly enable to downloading of batches of paper from the literature, extracting relevant information along with meta-data, and annotating the data. The tools are implemented in an open ecosystem that can be used from the command line or Python.

Strengths:

The tools developed here are really valuable for the future of large-scale analyses of the biomedical literature. This is a very well-written paper. The presentation of the use of the tools through several examples corresponding to different scientific questions really helps the readers to foresee the potential application of these tools.

Weaknesses:

The tools are command-based and store outcomes locally. So users who prefer to work only with GUI and web-based apps may have some difficulties. Furthermore, the outcomes of the tools are constrained by inherent limitations in the scientific literature, in particular, here the fact that only a small portion of the publications have full text openly available.

---

## [Referee Report · Reviewer #2 (Public Review)]

Summary:

In this manuscript, the authors described the litmining ecosystem that can flexibly combine automatic and manual annotation for meta-research.

Strengths:

Software development is crucial for cumulative science and of great value to the community. However, such works are often greatly under-valued in the current publish-or-perish research culture. Thus, I applaud the authors' efforts devoted to this project. All the tools and repositories are public and can be accessed or installed without difficulty. The results reported in the manuscript are also compelling that the ecosystem is relatively mature.

Weaknesses:

First and foremost, the logic flow of the current manuscript is difficult to follow.

The second issue is the results from the litmining ecosystem were not validated and the efficiency of using litmining was not quantified. To validate the results, it would be better to directly compare the results of litmining with recognized ground truth in each of the examples. To prove the efficiency of the current ecosystem, it would be better to use quantitative indices for comparing the litmining and the other two approaches (in terms of time and/or other costs in a typical meta-research).

The third family of issues is about the functionality of the litmining ecosystem. As the authors mentioned, the ecosystem can be used for multiple purposes, however, the description here is not sufficient for researchers to incorporate the litmining ecosystem into their meta-research project. Imagine that a group of researchers are interested in using the litmining ecosystem to facilitate their meta-analyses, how should they incorporate litmining into their workflow? I have this question because, in a complete meta-analysis, researchers are required to (1) search in more than one database to ensure the completeness of their literature search; (2) screen the articles from the searched articles, which requires inspection of the abstract and the pdf; (3) search all possible pdf file of included articles instead of only relying on the open-access pdf files on PMC database. That said, if researchers are interested in using litmining in a meta-analysis that follows reporting standards such as PRISMA, the following functionalities are crucial:

(a) How to incorporate the literature search results from different databases;

(b) After downloading the meta-data of articles from databases, how to identify whose pdf files can be downloaded from PMC and whose pdf files need to be searched from other resources;

(c) Is it possible to also annotate pdf files that were not downloaded by pubget?

(d) How to maintain and update the meta-data and intermediate data for a meta-analysis by using litmining? For example, after searching in a database using a specific command and conducting their meta-analysis, researchers may need to update the search results and include items after a certain period.

---

## [Referee Report · Reviewer #3 (Public Review)]

Summary:

The authors aimed to develop an automated tool to easily collect, process, and annotate the biomedical literature for higher efficiency and better reproducibility.

Strengths:

Two charms coming with the efforts made by the team are Pubget (for efficient and reliable grabbing articles from PubMed) and labelbuddy (for annotating text). They make text-mining of the biomedical literature more accessible, effective, and reproducible for streamlined text-mining and meta-science projects. The data were collected and analyzed using solid and validated methodology and demonstrated a very promising direction for meta-science studies.

Weaknesses:

More developments are needed for different resources of literature and strengths of AI-powered functions.

---

## [Author Response]

Thank you for the reviews and the eLife assessment. We want to take this opportunity to acknowledge the weaknesses pointed out by the reviewers and we will make small changes to the manuscript to account for these as part of the Version of Record.

**The tools are command-based and store outcomes locally**

We consider this to be an advantage of our ecosystem, which is intended for the case of individuals or small groups of authors. These features facilitate easy installation and integration with other tools. Further, our tool labelbuddy is a graphic user interface. Our tools may also be integrated into web-based systems as backends. Pubget is already being used in this way in the NeuroSynth Compose platform for semi-automated neuroimaging meta-analyses.

**pubget only gathers open-access papers from PubMed Central**

We recognize this as a limitation, and we acknowledge it in the original manuscript (in the discussion section, starting with "A limitation of Pubget is that it is restricted to the Open-Access subset of PMC"). We chose to limit the scope of our tools in order to ensure maintainability. Further, we are currently expanding pubget so it will also be able to access the abstracts and meta-data from closed-access papers indexed on PubMed. Future research could build other tools to work alongside pubget, to access other databases.

**Logic flow is difficult to follow**

We thank the reviewer for this feedback. Our paper describes an ecosystem of literature mining tools which does not lend itself to narrative flow nor does readily fit into the standard "Intro, Results, Discussion, Methods" structure that is typical in the scientific literature. We have done our best to conform to this expected format, but we have also provided detailed section and subsection headings to enable the reader to digest the paper nonlinearly. Each of the tools we describe also has detailed documentation on github that we update continuously.

**Results were not validated**

For the example where we automatically extracted participant demographics from papers, we validated the results on a held-out dataset of 100 manually-annotated papers. For the example with automatic large-scale meta-analyses (neuroquery and neurosynth), these methods are described together with their validation in the original papers. If this ecosystem of tools is integrated into other workflows, it should be validated in those contexts. We recognize that validating meta-analyses is a difficult problem because we do not have ground truth maps of the brain.

**Efficiency was not quantified**

Creators of tools do not always do experiments to quantify their efficiency and other qualities. We have chosen not to do this here, first because it is outside the scope of this paper as it would necessitate to specify very precise tasks and how efficiency is measured, and second because at least for the data collection part, the benefit of using an automated tool over manually downloading papers one by one is clear even without quantifying it. Compared to the approach of re-using existing datasets, our ecosystem is not necessarily more or less efficient. But it has other advantages, such as providing datasets that contain the latest literature, whereas the existing datasets are static and quickly out-of-date.

**We do not highlight the strength of AI functions**

We provide an example of using our tools to gather data and manually annotate a validation set for use with large language models (in our case, GPT). We are further exploring this domain in other projects; for example, for performing semi-automated meta-analyses using the NeuroSynth Compose platform. However, we did not deem it necessary to include more AI examples in the current paper; we only wanted to provide enough examples to demonstrate the scope of possible use cases of our ecosystem.

We thank the reviewers for their time and valuable feedback, which we will keep in mind in our future research.